# Study on Deterioration Characteristics of a Composite Crossarm Mandrel in a 10 kV Distribution Network Based on Multi-Factor Aging

**DOI:** 10.3390/polym15173576

**Published:** 2023-08-28

**Authors:** Long Ma, Xiaotao Fu, Lincong Chen, Xiaolin Chen, Cong Zhang, Xinran Li, Wei Li, Fangda Fu, Chuanfu Fu, Taobei Lin, Wensheng Mao, Hechen Liu

**Affiliations:** 1Key Laboratory of Physical and Chemical Analysis for Electric Power of Hainan Province, Hairui Road No. 23, Haikou 570100, China; fuxiaotao123abc@163.com (X.F.); 13907640264@163.com (L.C.); chenxiaol@hn.csg.cn (X.C.); zhangc7@hn.csg.cn (C.Z.); lixinran@hn.csg.cn (X.L.); 13703360546@163.com (W.L.); ffd@hn.csg.cn (F.F.); fuchuanfu@aliyun.com (C.F.); lintb@hn.csg.cn (T.L.); maows@hn.csg.cn (W.M.); 2Hebei Key Laboratory of Green and Efficient New Electrical Materials and Equipment, North China Electric Power University, Yonghua North Street No. 619, Baoding 071003, China; hc.liu@ncepu.edu.cn

**Keywords:** distribution network composite crossarm, composite mandrel, silicone rubber sheath, multi-factor aging system, mechanical properties, electrical insulation performance, microscopic analysis

## Abstract

This paper presents a study that conducted 5000 h of multi-factor aging tests on 10 kV composite crossarms, considering the natural environment in coastal areas and actual power line operations. Various aging conditions, such as voltage, rain, temperature, humidity, salt fog, ultraviolet light, and mechanical stress, were applied during the tests. The research initially analyzed the influence of multi-factor aging on the bending and tensile properties of the full-size composite crossarm. Subsequently, a detailed investigation was carried out to assess the impact of aging on the mechanical properties, electrical insulation properties, and microscopic characteristics of the composite crossarm core bar. Results indicated that the tensile strength and bending strength of the full-size composite crossarm mandrel experienced minimal changes after aging, remaining well within operational requirements. However, the silicone rubber outer sheath’s hydrophobicity decreased, leading to the appearance of cracks and holes on the surface, which provided pathways for moisture and salt infiltration into the mandrel. As a consequence, the bending strength and shear strength of the mandrel material were reduced by 16.5% and 37.7%, respectively. Moreover, the electrical performance test demonstrated a slight change in the mandrel’s leakage current, while the electrical breakdown strength decreased by 22.8%. Microscopic analysis using SEM, three-dimensional CT, and TGA revealed that a small amount of resin matrix decomposed and microcracks appeared on the surface. Additionally, the fiber-matrix interface experienced debonding and cracking, leading to an increased moisture absorption rate of the mandrel material.

## 1. Introduction

A composite crossarm is composed of a glass fiber-reinforced epoxy resin composite mandrel, a silicone rubber sheath, and fittings. It has the characteristics of good insulation performance, high strength, and light weight, which can effectively reduce the lightning trip-out rate of distribution lines in lightning-prone areas and improve the operation reliability of distribution network systems. In addition, compared with traditional iron crossarms, its manufacturing process saves a lot of steel, reduces the consumption of ore energy, and is more low-carbon and environmentally friendly. In 2016, the composite crossarm began to be applied on a pilot basis in some regions of China. By the end of 2020, China completed more than 30 pilot projects for composite crossarms [1]. The operating experience shows that the lightning trip rate of distribution network lines equipped with composite crossarms is greatly reduced. Figure 1 is the structural diagram of a composite crossarm.

The 10 kV composite crossarm running in China’s coastal areas is facing high temperature, high humidity, and high salt environmental tests. Although the mandrel is wrapped with a complete silicone rubber insulating sheath, with the increase in the operation time of the composite crossarm, the silicone rubber sheath is easy to damage and crack under the erosion of various environmental factors. The conductive particles in the atmosphere will diffuse into the material, causing agglomeration or hydrolysis of the mandrel resin matrix, fracture of the polymer molecular chain, debonding of the fiber/matrix interface, and finally, reducing the mechanical properties and insulation properties of the mandrel to varying degrees [2]. In addition, in coastal areas, under strong wind conditions, overhead conductors are prone to galloping; as a result, the composite crossarm bears mechanical fatigue stress for a long time and accelerates the cracking of mandrel material. Therefore, carrying out the accelerated aging test of the composite crossarm under multiple factors is of great significance for evaluating the service life of the composite crossarm and comprehensively analyzing the role and influence of various aging factors in the aging process of the composite crossarm.

Experts and researchers in China and abroad have long studied the aging characteristics of fiber-reinforced composite electrical equipment and its silicone rubber. Tian Liang et al. [3] studied the powder layer characteristics of composite insulator umbrella skirts aged for more than 10 years under three different environmental stresses. The results show that the outer sheath of a composite insulator has serious oxidative degradation under the action of various aging factors. Hou Sizu [4] and others found that ambient humidity will affect the temperature rise effect of a composite insulator, and its conductivity loss and polarization loss will increase with the increase in humidity. Nie Zhangxiang et al. [5] established the interface aging model of fiber-reinforced composites under the action of water and high temperature and evaluated the aging resistance of the material interface through the interface image aging rate and defect expansion index. Alsaadi et al. [6] explored the changes in mechanical properties of glass fiber/epoxy resin composites after damp heat and ultraviolet aging. The results showed that the interfacial shear strength decreased significantly after aging, and simultaneously, it was observed that the epoxy matrix of the composite was hydrolyzed. To sum up, in the past, the aging test and micro-detection of fiber composite materials were mainly aimed at suspension composite insulators, and most of them were single-factor aging, while the actual operating environment of the equipment was the joint action of multiple aging factors, which made the situation more complex. On the other hand, the mandrel of a composite insulator mainly bears tensile stress, while the mandrel of a composite crossarm mainly bears bending stress from the weight of the conductor. Because the bending strength of the composite perpendicular to the fiber direction is much less than the tensile strength parallel to the fiber direction, the composite crossarm is more vulnerable to fatigue damage by external mechanical stress.

In view of this, combined with the real operating environment of the composite crossarm, the 5000 h aging test of a distribution network’s composite insulated crossarm under multiple factors such as voltage, rain, temperature, humidity, salt fog, ultraviolet light, and mechanical fatigue is carried out in the laboratory. After aging, the outer sheath of the composite crossarm (hydrophobicity, surface micro-appearance) and the mandrel of the composite crossarm (mechanical properties, electrical insulation properties, physical and chemical properties) were analyzed by macro and micro tests, and the deterioration mechanism of the mandrel of the composite crossarm was explored.

## 2. Experiment

### 2.1. Materials

In this paper, the 10 kV composite crossarm (JZHD2-32 mm × 42 mm × 1750 mm) is provided by Jiangsu SHEMAR Power Co., Ltd. (Nantong City, China). The outer sheath and umbrella skirt are high-temperature vulcanized silicone rubber; the mandrel is based on bisphenol A epoxy resin and reinforced with glass fiber, in which glass fiber accounts for 80~85% of the total content.

### 2.2. Multi-Factor Aging Test of the Composite Crossarm

Referring to the IEC/TR 62730-2012 standard [7] and combined with the actual operation environment of composite crossarms, this paper establishes a set of test systems that can simultaneously carry out voltage, rain, temperature, humidity, mechanical force, salt spray, and UV aging tests of a 10 kV composite crossarm. Table 1 shows the system parameters of each aging factor, in which temperature and humidity, rainfall, and salt spray mainly simulate the natural climate of subtropical coastal areas. UV irradiation intensity refers to the GBT 16422.2 exposure test method of a plastic laboratory light source (method A), and its irradiation intensity is 550 W/m^2^. The mechanical load borne by the composite crossarm is calculated according to the DL/T 5220-2005 technical code for the design of overhead distribution lines of 10 kV and below. The maximum horizontal load and maximum bending load of the composite crossarm under natural operation come from strong wind environments and icing environments, which are 0.42 kN and 0.77 kN, respectively. This test takes 24 h as a cycle; each cycle is divided into 12 stages, each lasting two hours, for a total of 5000 h. Figure 2 shows the actual effect of the comprehensive aging system.

### 2.3. Hydrophobic Property Test of Outer Sheath of Composite Crossarm after Aging

In this paper, the static contact angle method (CA method) is used to measure the hydrophobic angle of the silicone rubber outer sheath before and after aging [8]. Two 3 cm × 3 cm × 5 mm outer sheath samples were cut between each of the two umbrella skirts on the aging composite crossarm, for a total of 24 pieces. After testing the static contact angle of 5 water droplets for each sample, take the average value.

### 2.4. Test of Mechanical Properties of the Full-Size Composite Crossarm

In accordance with the standard Q/GDW 12069-2020 [9], titled “Technical Specification for Composite Insulation Crossarm for 10 kV Distribution Lines”, a comprehensive assessment of the composite crossarm is performed. This evaluation involves conducting tests to determine the bending and tensile properties of the crossarm, closely observing its stress and deformation characteristics. The primary objective is to ascertain whether the crossarm remains in compliance with operational requirements and can continue to function effectively.

Tensile test: fix the composite crossarm on the reaction support through the connecting plate and load it reversely at the fixed fittings. The load should be smoothly increased to 10 kN within 30~90 s. If the mandrel is not pulled out from the end fittings, the test is qualified. Bending test: the bending load failure test is carried out on the aged composite crossarm by using the bending torsion test machine, which gradually increases to the failure of the mandrel or end accessories within 30~300 s. The maximum load measured in the test is regarded as the bending failure load. The mechanical strength test is carried out at a temperature of 20 ± 10 °C. The test arrangement is shown in Figure 3.

### 2.5. Performance Test of the Composite Crossarm Mandrel Material

The full-scale mechanical test can only reflect the overall mechanical strength of the composite crossarm but cannot accurately reflect the relevant properties of its mandrel material. Therefore, this paper further carries out an experimental analysis of the material properties of the composite crossarm mandrel before and after aging. In Figure 1, corebar samples are taken from two locations of the composite crossarm: the root (near the fixed hardware end) and the end (near the wire suspension end). These samples are labeled S-root and S-terminal, respectively. The unaged sample is labeled as S-unaged.

(1)Mechanical test

The composite crossarm mainly bears the bending load during operation, so the bending test and interlaminar shear test analysis are mainly carried out for the mandrel material. Bending test: according to the requirements of ISO 178-2010, the sample size is 40 mm × 15 mm × 2 mm, the span is 32 mm, and the load loading rate is 2 mm/min. Shear test: according to the requirements of standard ISO 14130-1997 [10], the size is designed as 20 mm × 20 mm × 1 mm, the span is 5 mm, and the load loading rate is the same as that of the bending test. To ensure the precision of the results, the average value was calculated for each type of sample using 10 sets of data.

(2)Moisture absorption rate

The size of the mandrel water absorption sample is 30 mm × 40 mm × 1 mm. Prior to conducting the moisture absorption test, the samples undergo drying pretreatment. They are dried in a vacuum drying oven at 100 °C for 1 h. The mass of the dried sample is then measured using a photoelectric analytical balance, and this recorded mass is denoted as *m*_0_. Then, the dried sample is soaked in deionized water; the sample is taken out at certain times, wiped dry within 20 s, the mass after *t* hours of moisture absorption is recorded as *m_t_*. The moisture absorption rate, *ω,* of each sample during the experiment is calculated using the following formula:
(1)$$\omega =\frac{m_{t}-m_{0}}{m_{0}}\times 100\%$$

(3)Electrical test

Leakage current test: The test voltage ranges from 0 to 12 kV, with a boost rate of 2 kV/s. A test sample, measuring 25 to 30 mm in thickness, is extracted from the aged composite crossarm. The objective is to measure the leakage current within the mandrel sample. Insulation strength: According to the requirements of the IEC 60243-1-2013 standard [11], the test is carried out with a spherical electrode, the boost rate is 2 kV/s, and the sample size is 30 mm × 40 mm × 1 mm. In order to prevent surface flashover during the test, immerse the sample and electrode in dimethyl silicone oil. At least 15 valid data points are obtained for each sample, and the experimental results are statistically analyzed by the Weibull distribution.

(4)TGA

Thermal stability is an important factor in measuring the aging resistance of mandrel materials. The thermal weight loss was tested by a TG/DTA thermogravimetric analyzer. Test conditions: the mandrel sample was placed in Al_2_O_3_ crucible and tested in an air atmosphere at a heating rate of 10 °C/min. The test temperature range is 35~800 °C.

(5)SEM analysis and 3D CT nondestructive testing

In order to observe the deterioration degree of the silicone rubber outer sheath and built-in mandrel on the surface of the composite crossarm, a Nova NanoSEM450 scanning electron microscope and a nanovoxel-3000 open-tube transmission high-resolution CT system were used to carry out micro-nondestructive testing on the aged composite crossarm, which truly reflects the internal structure of the sample.

## 3. Analysis and Discussion of Experimental Results

### 3.1. Mechanical Properties Test of the Full-Size Composite Crossarm

The appearance of the composite crossarm after the tensile test is shown in Figure 4. It can be seen from the figure that after aging under the horizontal tension of 10 kN, the mandrel prolapse and shed cracking did not occur in the composite crossarm, and the metal fittings and the mandrel are in good condition after long-term multi-factor aging on the surface. This shows that the composite crossarm has good anti-aging performance, and both the inner glass fiber composite core rod and the outer silicone rubber sheath have very reliable performance after aging. Since the maximum horizontal load of the composite insulating crossarm is only 0.42 kN under strong wind conditions, the tensile performance of the composite insulating crossarm after aging far exceeds the level required for normal operation, which is the key to its wide application in practical applications.

Figure 5 shows the load variation curve of a full-size composite crossarm bending failure test. It can be seen from the figure that the maximum bending load that the composite crossarm can bear decreases after multi-factor aging. The maximum bending force of the unaged mandrel is 13.3 kN, and the average maximum bending force of the aged mandrel is 12.27 kN, a decrease of 7.74%. The maximum bending load of the composite crossarm is 0.77 kN under ice conditions, which is far less than the bending failure load of the rod after aging. To sum up, although the bending resistance of the aged composite crossarm mandrel has declined, overall it still meets the mechanical strength requirements.

The full-size tensile test and bending test of the composite crossarm can evaluate whether it meets the operation requirements, but they cannot reflect whether there are small defects at different positions of the composite crossarm mandrel material. Therefore, it is necessary to intercept samples from the outer sheath and mandrel of the aged composite crossarm to further explore the internal damage of the material and analyze the influence of various aging factors.

### 3.2. Hydrophobic Characteristics of the Outer Sheath of the Composite Crossarm after Aging

The distribution of average hydrophobic angle data for the silicone rubber outer sheath of the composite crossarm is presented in Figure 6. The findings indicate that the unaged composite crossarm exhibits excellent initial hydrophobicity, with 95.8% of the hydrophobic angle readings exceeding 100°. In contrast, the aging crossarm shows a decrease in hydrophobicity of only 76.7%, which indicates that the water-repellent performance of the outer sheath has declined to a certain extent after aging, but it still retains good water-repellent ability. This is considered to be because the silicone rubber material of the outer sheath deteriorates and its hydrophobicity decreases after being subjected to high temperatures, high humidity, salt spray corrosion, and ultraviolet radiation. A Nova NanoSEM450 scanning electron microscope was used to observe the microscopic morphology of the silicone rubber outer sheath and analyze the degree of surface deterioration.

In Figure 7a, the surface of the outer sheath of the unaged composite crossarm is flat without obvious holes or gaps. Figure 7b,c shows the outer sheath of the aged composite crossarm. Compared with the unaged composite crossarm, there are long cracks and holes with rough boundaries and clear centers on the surface of the outer sheath, with a size of about 10.24 μm. The holes’ shape is concave, and the aging shows a trend of developing to the inside of the material. It can be seen that aging factors such as salt spray corrosion, alternation of damp and heat, UV irradiation, and mechanical fatigue will degrade the outer sheath of the composite crossarm and degrade the overall hydrophobic performance. At the same time, during aging, H_2_O, Na^+^, and Cl^−^ can penetrate into the composite crossarm with the help of holes and cracks on the surface of the outer sheath, eroding the mandrel of the composite crossarm, which may degrade the mechanical and electrical properties of the mandrel material and lay hidden dangers for the safe operation of the composite crossarm. Therefore, it is necessary to further study the performance indexes of the aged composite crossarm mandrel.

### 3.3. Macroscopic Characteristics of the Composite Crossarm Mandrel

(1)Mechanical test

Figure 8 shows the experimental results of the mechanical properties of the composite crossarm mandrel material, and the shaded part of the figure represents the distribution of the experimental data. With the increase in aging time, the bending properties and interlaminar shear properties of mandrel materials from different positions of the composite crossarm decreased to a certain extent; the average bending strength and average shear strength of S-root decreased by 15.2% and 38.2%, respectively, while those of S-terminal decreased by 5.1% and 28.9%, respectively. The decrease in bending properties and interlaminar shear properties of the sample indicates that the fiber mechanical strength of the mandrel material, the filling state of the resin matrix, and the bonding strength of the resin/fiber interface may have changed under multi-factor aging.

In addition, the decrease in interlaminar shear strength is more significant than that of flexural strength in both S-root and S-terminal. This is because the shear plane of the shear test is parallel to the fiber direction, and the test mainly reflects the bonding strength of the mandrel fiber/resin matrix interface, while the force application direction of the bending test is perpendicular to the fiber direction, and the change range is relatively small due to the joint action of resin and fiber. The sharp decline in interlaminar shear properties indicates that multi-factor aging may have an impact on the interfacial bonding strength of composite crossarm fiber/resin. The fiber/resin interface is the weak link in the mandrel material. When the mandrel is affected by rain, salt spray, and high temperatures and the external moisture invades the mandrel through the damaged outer sheath, the fiber and resin may expand differently [12], resulting in interface debonding and affecting the mechanical properties of the material. At the same time, the aging environment of continuous high temperature and high humidity will also increase the plasticity of the resin matrix and reduce the stiffness of the material [13]. The macroscopic performance is that the mechanical strength of the mandrel material decreases.

When comparing the mechanical properties of the composite crossarm mandrel materials at different positions, it is found that the decline in the mechanical properties of S-root is more obvious than that of S-terminal. In this paper, the SolidWorks simulation finite element simulation software is used to establish the simulation model and analyze the stress distribution of the composite crossarm according to the actual size and applied load (horizontal load: 0.18 kN, vertical load: 0.32 kN) of the 10 kV composite crossarm in the multi-factor aging experiment. The simulation results are shown in Figure 9. The bending stress at the root of the composite crossarm mandrel is the largest, at about 23.63 MPa, while the end of the composite crossarm is only 1.969 MPa. Because the mechanical stress borne by the root of the composite crossarm mandrel is higher than that at the end, more small defects may be generated in it under the action of long-term mechanical stress, resulting in the decline in its mechanical properties.

(2)Moisture absorption rate

By fitting the moisture absorption test data of the composite crossarm mandrel based on the Fick diffusion model [14], the diffusion coefficient of each sample is calculated. The moisture absorption rate ($c_{t}$) of each sample during the experiment is calculated as shown in Formula (2).
(2)$$c_{t}=\frac{m_{t}-m_{0}}{m_{0}}\times 100\%$$

In Formula (2), $c_{t}$ is the moisture absorption rate of the sample at time *t*, $m_{t}$ is the mass of the sample at time *t*, and $m_{0}$ is the initial mass of the sample.

The Fick diffusion theory is the basic theory for studying the molecular diffusion process. In this study, the Fick diffusion model is used to analyze the moisture absorption behavior of composite insulator core rods.
(3)$$\frac{c_{t}}{c_{\infty }}=1-\frac{8}{\pi^{2}}\overset{\infty }{\underset{n=0}{\sum }}\frac{{\left(-1\right)}^{n}}{{\left(2n+1\right)}^{2}}exp\left[-\pi^{2}{\left(2n+1\right)}^{2}\frac{D\cdot t}{h^{2}}\right]$$

In Formula (3), D is the diffusion coefficient of water molecules, and h is the thickness of the mandrel sample. By fitting the experimental data, the expression of the Fick diffusion model can be obtained in Figure 10.

Within 300 h, the moisture absorption rate of all samples increased with the increase in moisture absorption time and gradually stabilized. The moisture absorption rate of the aged mandrel is higher than that of the unaged mandrel, in which the diffusion coefficient of the aged sample S-root is the largest and the moisture absorption speed is the fastest, indicating that multi-factor aging does affect the internal structure of the composite crossarm mandrel, resulting in the enhancement of its moisture absorption capacity. Considering that after the continuous aging of the mandrel material under mechanical force, some resin matrix forms microcracks, and the resin matrix is the main body of the mandrel material to absorb water, in the Humid Aging environment, a small amount of water enters the resin through diffusion and “capillary action” [15], causing the expansion and crack propagation of the resin matrix, thus increasing the saturated moisture absorption rate. On the other hand, the hygroscopic expansion of the matrix will also affect the adhesion of the fiber/matrix interface. The interface bonding failure in some areas forms pores, which increase the moisture absorption capacity of the mandrel. Moreover, when some fibers are exposed due to interfacial debonding, due to the large relative surface area of glass fibers, water is easy to adhere to the surface, which will also increase the moisture absorption rate of the mandrel material.

(3)Electrical performance test and 3D CT scanning

Figure 11a shows the change in internal leakage current of the mandrel with the increase in test voltage before and after sample aging. After comparing the data in the figure, it is found that there is little difference in the amplitude of leakage current between the aged mandrel sample and the unaged mandrel under each test voltage. Further observation shows that the leakage current of S-root is slightly greater than that of S-terminal under the same test voltage. According to the moisture absorption parameters of the Fick diffusion model, the diffusion coefficient of S-root is greater than that of S-terminal. Therefore, conductive particles such as H_2_O, Na^+^, and Cl^−^ diffuse faster in S-root, which increases the conductivity of mandrel material, resulting in a faster rise in leakage current. However, overall, the leakage current of the aged composite crossarm mandrel material increases slightly below the 100 μA required for operation, and good insulation is still maintained.

The power frequency breakdown voltage test and three-dimensional microscopic scanning were carried out on the samples before and after aging. The Weibull distribution curve of mandrel breakdown field strength and three-dimensional scanning results are shown in Figure 11b,c. The power frequency breakdown field strength of the sample S-unaged with 63.2% breakdown probability is 21.9 kV/mm, while the breakdown field strength of sample S-root is 16.89 kV/mm; S-terminal is 18.84 kV/mm, and the breakdown strength of S-root is 22.8% lower than that of S-unaged. According to the above test results, it is clear that the integrity of the internal structure of the mandrel material has an impact on its breakdown strength. Observing the three-dimensional image of the mandrel, it is found that there are no obvious micro-defects in the unaged mandrel, which plays the role of a uniform electric field to a certain extent, so its breakdown field strength is high. After aging, small pores appear in some areas of the mandrel, and some pores are relatively concentrated. With the trend of mutual connection, when the applied voltage is high enough, the internal defects of the mandrel will produce a partial discharge, which greatly promotes the process of breakdown. On the other hand, in view of the little change in the data before and after the aging of the mandrel in the leakage current test, combined with the three-dimensional scanning results of the mandrel, it is considered that this is because the voltage applied when testing the leakage current of the mandrel material is low, and the small defects in the mandrel do not reach the through discharge, so most of the mandrel still shows good insulation.

### 3.4. Micro-Properties of the Composite Crossarm Mandrel

(1)SEM

The SEM results of the composite crossarm mandrel before and after aging are shown in Figure 12. As can be seen from Figure 12a, the glass fibers of the unaged sample are arranged in order and parallel, the glass fibers are completely covered by epoxy resin, and the interface is well bonded. Figure 12b shows a sample of the S-terminal mandrel. There is a small amount of decomposition of the resin on the surface, but the glass fibers are still closely arranged, and the residue generated by resin decomposition is attached to the fiber surface. Compared with S-terminal, the fibers of S-root in Figure 12c show an uneven brush shape, some fiber orientations change, and the arrangement at the fracture is slightly disordered. A small amount of decomposition of resin will lose the connection effect on some glass fibers, and the aggregation degree of glass fibers will decrease accordingly. This result is basically consistent with the wet and heat aging characteristics of mandrels reported in the literature [16].

According to the SEM observations, the resin matrix of the mandrel is degraded, and microcracks and internal pores appear during the aging process, which will broaden the contact area between water molecules and materials and increase the free volume of stored water molecules. As moisture seeps into the mandrel during the aging cycle, the aging speed will continue to accelerate. At the same time, due to the effect of mechanical fatigue, the crack will also extend to the interior of the sample, resulting in the interface bonding failure of some mandrel composites. In addition, the thermal expansion coefficients of resin matrix and glass fiber at 50 °C aging temperatures are different (resin matrix: 56.8 × 10^−6^ m/°C, glass fiber: 4.8 × 10^−6^ m/°C). When facing aging factors such as rain and high temperatures, a certain thermal strain will be generated inside the mandrel material, which will further promote the debonding of the resin/fiber interface.

(2)TGA

As can be seen from the thermo-gravimetric curves in Figure 13a, the mass loss of the mandrel samples before and after aging starts around 300 °C. Figure 13b shows the thermal weight loss rate curve. There is a weight loss rate peak before and after mandrel aging, and the temperature range is 380 °C~460 °C. The weight loss rate of the sample first increases and then decreases with the increase in temperature, reaching a maximum value of 405~415 °C. Since the melting point of glass fiber is about 1300 °C, which is much higher than this temperature range, and the decomposition starting temperature of epoxy resin is just about 300 °C, it is preliminarily judged that this stage is mainly caused by the pyrolysis of the epoxy resin matrix in the mandrel sample. In Figure 13a, the mass percentage of epoxy resin in the unaged sample is maintained at about 13.68%, and the aged mandrel sample is 11.53%. The aged mandrel resin matrix is slightly degraded, but the overall difference is small.

In this paper, the empirical formula of reaction rate varying with temperature, the Arrhenius equation, and the reaction kinetic model are used to calculate the activation energy of materials before and after aging [17,18,19,20]. Activation energy is the minimum energy required for the chemical reaction, and its value represents the difficulty of the pyrolysis reaction of the mandrel sample.

*α*(*T*) is the degree of conversion at temperature *T*, and its value can be obtained from the TGA data of the sample according to Equation (4). In the equation, $\omega_{0}$ is the initial weight, $\omega_{f}$ is the final weight, and $\omega_{T}$ is the weight at temperature *T*.
(4)$$\alpha \left(T\right)=\frac{\omega_{0}-\omega_{T}}{\omega_{0}-\omega_{f}}$$

DEAM is usually used to analyze complex reactions such as biomass and fossil fuel pyrolysis. When the total volatile content changes during non-isothermal pyrolysis, the model is shown in the formula. In this equation, $k_{0}$ is the pre-exponential factor corresponding to the E value, β is the heating rate, R is the universal gas constant, T is the temperature, and $f\left(E\right)$ is the distribution of the activation energy, representing the differences in the activation energies of many reactions.
(5)$$\alpha \left(T\right)=\int_{0}^{\infty }1-exp\left\{\left[-\frac{k_{0}}{\beta }\int_{0}^{T}exp\left(-\frac{E}{RT}\right)\right]\right\}dTf\left(E\right)dE$$

In order to estimate the values of $k_{0}$ and $f\left(E\right)$, the activation energy distribution is generally assumed by a Gaussian distribution with mean activation energy $E_{0}$ and standard deviation σ, as shown in Equation (6).
(6)$$f\left(E\right)=\frac{1}{\sigma \sqrt{2\pi }}exp\left[-\frac{{\left(E-E_{0}\right)}^{2}}{2\sigma^{2}}\right]$$

The variation of the pyrolysis activation energy of the mandrel sample with temperature is shown in Figure 13c. On the whole, the activation energy first increases and then decreases with the increase in temperature and reaches a maximum at about 415 °C, which corresponds to the weight loss rate peak in the DTG curve. On the other hand, comparing the change trend of activation energy before and after mandrel aging, it is found that the activation energy of the aged mandrel is less than that of the unaged mandrel at the same temperature within 380 °C~415 °C, indicating that the aged mandrel is more prone to pyrolysis in this temperature range, that is, the thermal stability of the aged mandrel is weaker than that of the unaged mandrel. Combined with the macro test results and micro appearance characteristics of the mandrel sample, it can be seen that the outer sheath of the composite crossarm has cracks and holes after multi-factor aging. In the rain and salt fog aging cycle, when the moisture accelerates to invade the interior of the composite crossarm, the mandrel material will suffer from damp heat aging, in which a small amount of epoxy resin main chain will break [21]. The bond energy of the main chain of the resin is high, and the activation energy required for the reaction is also high. The aging mandrel without part of the main chain can undergo pyrolysis reactions of small molecules such as aldehydes and ketones without high activation energy. Therefore, the thermal stability of the aging mandrel decreases.

### 3.5. Multi-Factor Aging Mechanism Analysis of the Composite Crossarm

During the aging process of the 10 kV composite crossarm, it is subject to numerous influencing factors, including voltage fluctuations, exposure to rain, high temperatures, humidity, salt fog, ultraviolet light, and mechanical fatigue. Figure 14 is the schematic diagram of the multi-factor aging mechanism of the composite crossarm. The effects of various factors on the mandrel and outer sheath of the composite crossarm include the following:(1)Outer sheath aging. The composite crossarm worked under multi-factor aging for 5000 h. The main chain of Si-O-Si in the material broke due to rain, high temperatures, salt spray corrosion, and UV irradiation. The hydrophobic group Si-CH_3_ decreased and the hydrophilic group OH increased [22,23], resulting in a decrease in the hydrophobicity of the silicone rubber sheath. At the same time, the mechanical fatigue vibration will also aggravate the cracking of the outer sheath. According to the results of SEM, there are cracks and holes on the surface of silicone rubber, which provide a diffusion channel for H_2_O, Na^+^, and Cl^−^ to invade the mandrel.(2)Mandrel material aging. In terms of the resin matrix, TG test results show that after H_2_O, Na^+^, and Cl^−^ penetrate into the mandrel through the damaged outer sheath, a small amount of the resin matrix decomposes and the molecular chain breaks under the combined aging of rain and high temperature, resulting in a decline in its thermal stability. At the same time, the continuously humid and hot environment will also increase the plasticity and reduce the stiffness of the resin matrix. For the fiber/resin interface, moisture and other ions infiltrating into the mandrel will cause uneven expansion in the humid and hot environment, resulting in interfacial debonding. From the results of three-dimensional CT and SEM, under the long-term action of high temperature, high humidity, and high salt accompanied by mechanical load, there are microcracks and pores in the resin matrix and fiber/resin interface, and local defects have a trend toward connectivity. This will affect the macroscopic mechanical properties, electrical properties, and moisture absorption characteristics of the core shaft material to a certain extent.

## 4. Conclusions

In this study, we conducted aging tests on 10 kV composite crossarms under various environmental conditions, including voltage, rain, temperature, humidity, salt spray, ultraviolet light, and mechanical load. We compared the macroscopic and microscopic characteristics of the outer sheath and mandrel before and after aging and evaluated the degree of deterioration of the 10 kV composite crossarm mandrel after long-term operation. The following conclusions were drawn:(1)The tensile strength and bending strength of the full-scale composite crossarm after aging decreased by only 7.74%, still well above the rated load during normal operation, meeting operational requirements. However, the overall mechanical strength design margin of the composite crossarm is relatively large. In areas where the natural environment aligns with the multi-factor aging system parameters in this study, a moderate reduction in the margin is feasible.(2)After aging, the hydrophobicity of the composite crossarm’s outer sheath decreases, leading to the appearance of cracks and holes on the surface. This allows a small amount of moisture from the outside to penetrate into the mandrel through the sheath. For coastal environments, it is crucial to focus on enhancing the quality of the outer sheath to effectively protect the internal mandrel.(3)The test results of mandrel materials suggest that improving the bonding strength and thermal stability of the fiber-resin interface can help reduce the formation of defects, such as internal pores in the crossarm, in harsh environments. This improvement contributes to enhancing the mechanical and electrical strength of the composite crossarm.

These findings underscore the significance of considering aging effects and environmental conditions when designing and utilizing composite crossarms, especially in coastal applications. Our study provides valuable insights into the performance of composite crossarm materials in real-world applications, aiding in making informed decisions in power transmission and distribution systems.

## Figures and Tables

**Figure 1 polymers-15-03576-f001:**
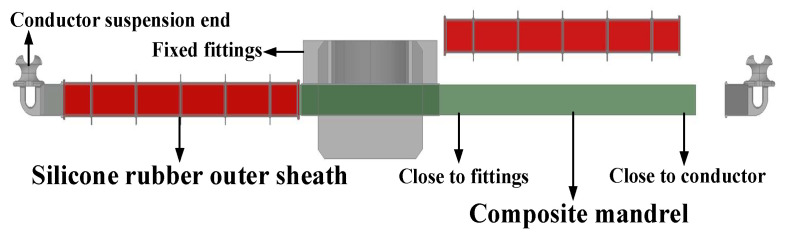
Schematic diagram of a composite crossarm of the distribution network.

**Figure 2 polymers-15-03576-f002:**
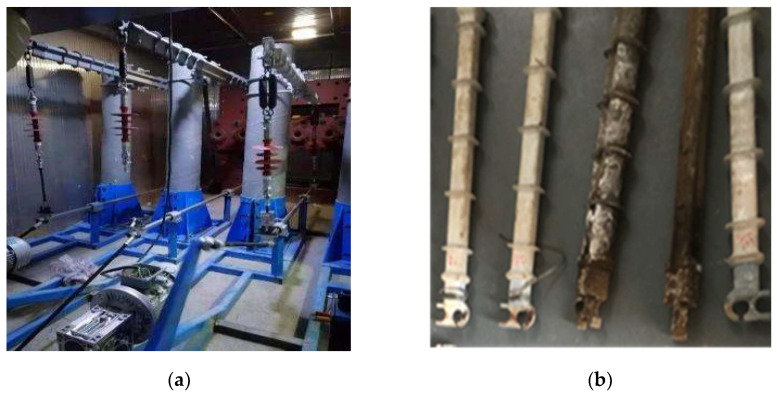
Actual effect drawing of a multi-factor aging system. (**a**) General view of set-up. (**b**) Crossarm after aging.

**Figure 3 polymers-15-03576-f003:**
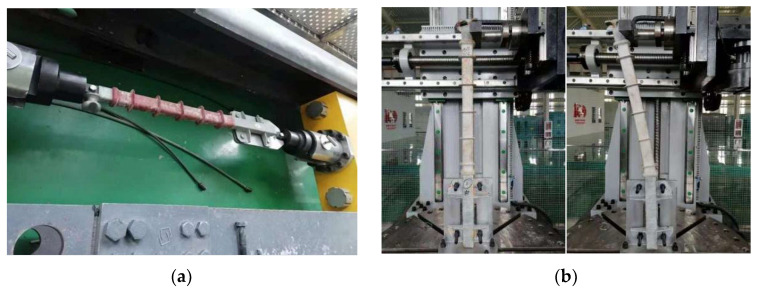
Mechanical test of a full-size composite crossarm. (**a**) Tensile strength test. (**b**) Bending strength test (for (**b**), the photo on the left is the crossarm before the bend test started, and the photo on the right is the crossarm during the bend test).

**Figure 4 polymers-15-03576-f004:**
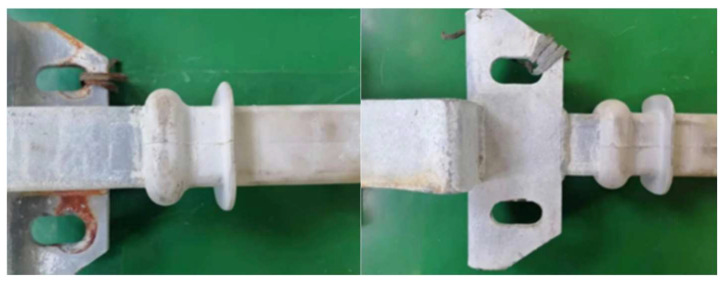
Appearance of the composite crossarm after the tensile test.

**Figure 5 polymers-15-03576-f005:**
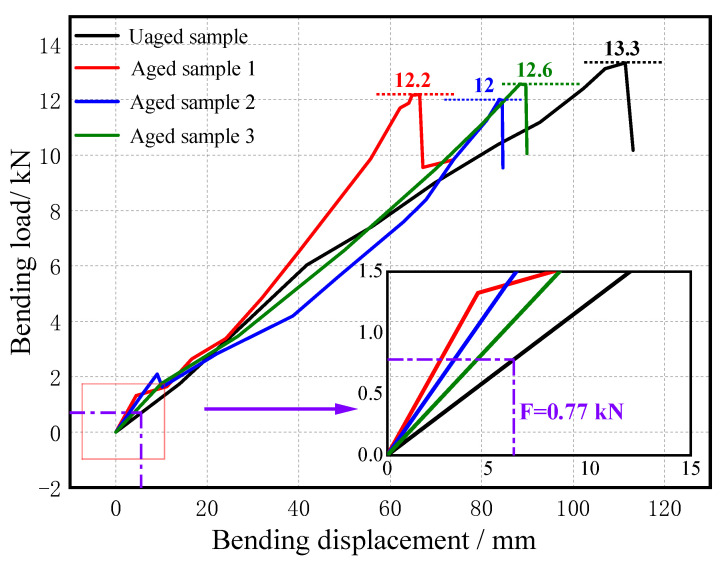
Load variation curve of the full-scale composite crossarm bending failure test.

**Figure 6 polymers-15-03576-f006:**
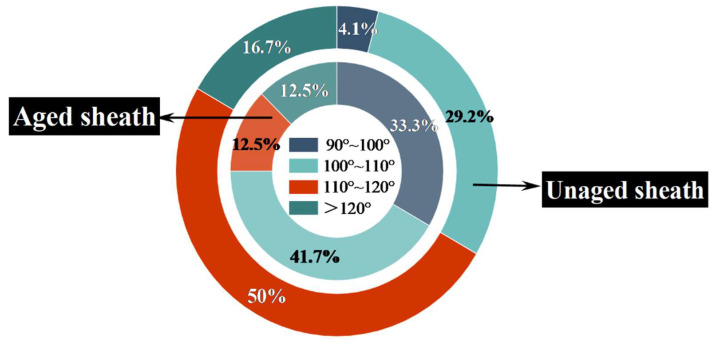
Static hydrophobic angle distribution of the silicone rubber sheath.

**Figure 7 polymers-15-03576-f007:**
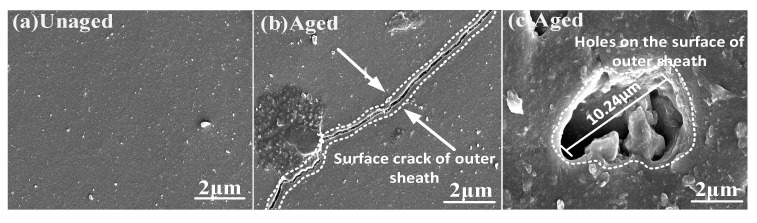
Scanning electron microscopy images of the surface of the silicone rubber outer sheath.

**Figure 8 polymers-15-03576-f008:**
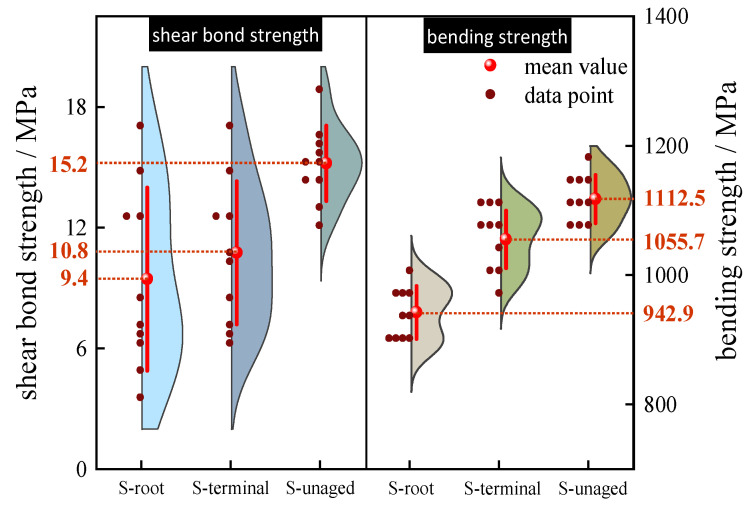
Data distribution of the interlaminar shear strength and flexural strength of the composite crossarm mandrel.

**Figure 9 polymers-15-03576-f009:**
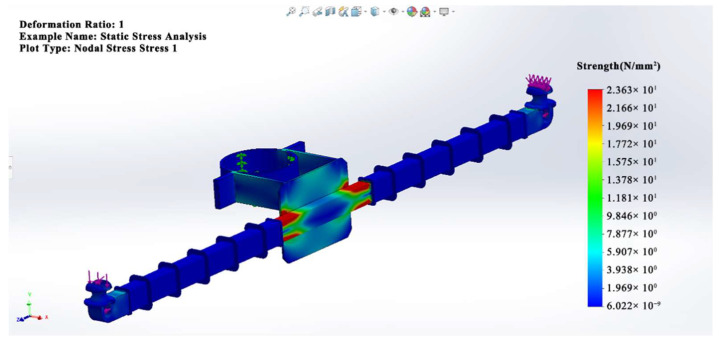
Simulation diagram of the bending strength of the composite crossarm under natural operating conditions.

**Figure 10 polymers-15-03576-f010:**
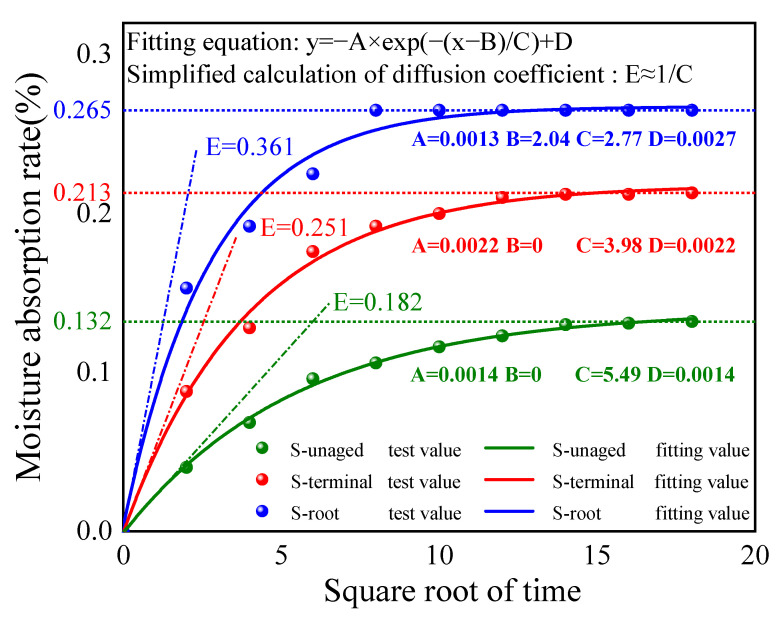
Variation curve of the moisture absorption rate of the composite crossarm mandrel with the square root of time.

**Figure 11 polymers-15-03576-f011:**
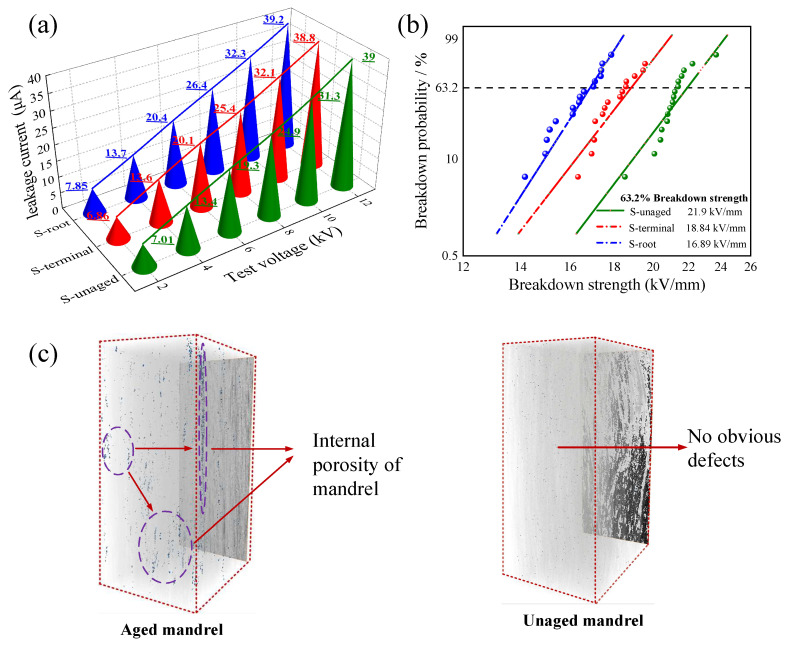
(**a**) Leakage current variation diagram; (**b**) breakdown strength variation diagram; (**c**) three-dimensional scanning image.

**Figure 12 polymers-15-03576-f012:**
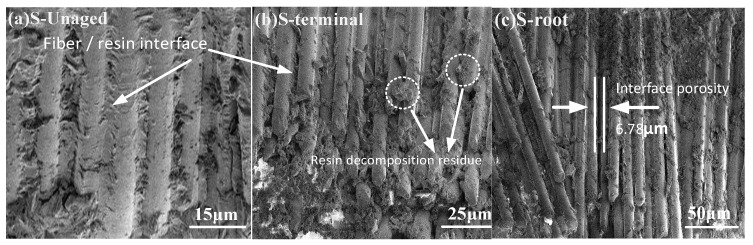
SEM results of the mandrel material before and after aging.

**Figure 13 polymers-15-03576-f013:**
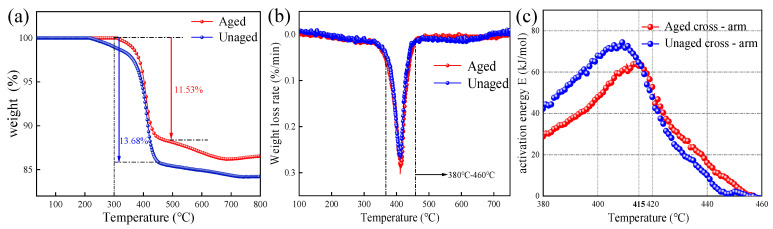
(**a**) TG curve of the mandrel material; (**b**) DTG curve of the mandrel material; (**c**) activation energy curve of mandrel pyrolysis.

**Figure 14 polymers-15-03576-f014:**
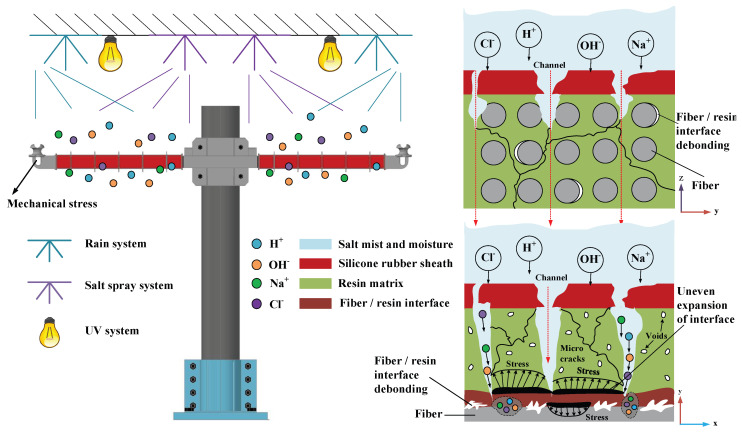
Schematic diagram of multi-factor aging of the composite crossarm mandrel.

**Table 1 polymers-15-03576-t001:** Multi-factor aging subsystem design parameters.

Aging Factor	Parametric Design
Humidity	98% RH
Temperature	50 °C ± 0.5 °C
Rain wet	24 h rainfall 50~100 mm
Salt spray	Particle size: 5~10 μm; NaCl content: 7 kg/m^3^; flow rate: 0.5 kg/m^3^ h
UV	Xenon lamp power: 6 kW; UV wavelength: 290~800 nm; irradiance: 550 W/m^2^
Mechanics	Horizontal load: 177.89 N; vertical load: 319.69 N; Mechanical force frequency: 0.6 Hz
Voltage	10 kV

## Data Availability

Not applicable.

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
