# Peer review of "Study on Deterioration Characteristics of a Composite Crossarm Mandrel in a 10 kV Distribution Network Based on Multi-Factor Aging"

_polymers, 2023, doi:10.3390/polym15173576_

Round 1

Reviewer 1 Report

The topic of the manuscript is interesting from the practical point of view in the field of understanding deterioration on the characteristics of a composite arm mandrel. 

The paper requires careful language polishing from the abstract to the conclusion. Check all the abbreviations used, particularly prior to using them, as well as capitalization

The keywords used could be improved in accordance with the manuscript and the title.

The title corresponds to the purpose of the article.

The abstract requires major revision, information is not well presented. It is difficult to follow the main idea of the manuscript. 

Experimental details are not completely clear. Improve organization and order. 

Results are presented but not completely discussed. Authors are encouraged to carry out a deep analysis of the results. For instance, the authors found out that after the tensile test (Figure 4) of the aged sample, there was no detachment and umbrella skirt cracking, but there is no fundamental point of view of why? Is it a common behavior? Expected? Please, explain in detail and improve the information in the manuscript.

Figure 9 is not clear, it is important to include the text in English. Improve the quality of the figure.

The results from the moisture absorption rate are really difficult to follow. The obvious is presented, but not the proper mechanisms that take place. Why do the authors mention capillary action between quotation marks?

Why the authors referred to H2O, Na+, and Cl- as conductive particles (see page 10 line 320)?

Conclusions are not concise. A strengthened conclusion is required to highlight the results' importance rather than a list of the results.

English must be improved and carefully checked.

Author Response

Thank you for your valuable comments, the point-by-point response will be uploaded as a Word file. See the attachment please.

Reviewer 2 Report

In this manuscript, the authors studied the aging test of a 10kV Composite cross arm under the conditions of voltage, rain, temperature, humidity, salt fog, ultraviolet light, and mechanical load is carried out. They compared the macro and micro characteristics of the outer sheath and mandrel before and after the aging of the composite cross arm. The research shows that the tensile strength and bending strength of the full-size composite cross arm mandrel after aging have little change, which all meet the operation requirements. The hydrophobicity of the silicone rubber outer sheath of the composite cross arm decreases, and cracks and holes appear on the surface, providing a channel for moisture and salt to invade the mandrel. The fiber-matrix interface was debonding and cracked, and the moisture absorption rate of the mandrel material increased. The research conclusion has an important reference value for evaluating the operation stability of the 10kV Composite cross arm and revealing the multi-factor deterioration mechanism of the composite mandrel. The manuscript should include a clear experimental setup and data analysis section. The interpretations of the results were deeply discussed. The quantity and quality of the figures are appropriate. We believe that this research subject is promising for studying the aging characteristics of fiber-reinforced composite electrical equipment and its silicone rubber. The aging test and micro detection of fiber composite materials were mainly aimed at suspension composite insulators, and most of them were single factor aging, while the actual operating environment of the equipment was the joint action of multiple aging factors, which made the situation more complex.

My comments as;

1)      The comma should be added in the title a “…..field, and temperature…….”

2)      Line 140, “The mechanical strength test is carried 140 out at the temperature of 20℃± 10K”. the unit of the temperature should be the same.

3)      Line 291, “Fit the moisture absorption test data of composite cross arm mandrel based on Fick diffusion model [10], and calculate the diffusion coefficient of each sample, The results are shown in Figure 10.” The parameters in Fig. 10 should be defined and discussed.

4)      Line 292, “and calculate the diffusion coefficient of each sample, The results are shown in Figure 10”. This sentence needs to be revised.

5)      Line 387, “In this paper, the empirical formula of reaction rate varying with temperature, Arrhenius equation and reaction kinetic model are used to calculate the activation energy of materials before and after aging [13-16].” The way for calculating the activation energy should be included in the text.

Line 481, Page number?

In this manuscript, the authors studied the aging test of a 10kV Composite cross arm under the conditions of voltage, rain, temperature, humidity, salt fog, ultraviolet light, and mechanical load is carried out. They compared the macro and micro characteristics of the outer sheath and mandrel before and after the aging of the composite cross arm. The research shows that the tensile strength and bending strength of the full-size composite cross arm mandrel after aging have little change, which all meet the operation requirements. The hydrophobicity of the silicone rubber outer sheath of the composite cross arm decreases, and cracks and holes appear on the surface, providing a channel for moisture and salt to invade the mandrel. The fiber-matrix interface was debonding and cracked, and the moisture absorption rate of the mandrel material increased. The research conclusion has an important reference value for evaluating the operation stability of the 10kV Composite cross arm and revealing the multi-factor deterioration mechanism of the composite mandrel. The manuscript should include a clear experimental setup and data analysis section. The interpretations of the results were deeply discussed. The quantity and quality of the figures are appropriate. We believe that this research subject is promising for studying the aging characteristics of fiber-reinforced composite electrical equipment and its silicone rubber. The aging test and micro detection of fiber composite materials were mainly aimed at suspension composite insulators, and most of them were single factor aging, while the actual operating environment of the equipment was the joint action of multiple aging factors, which made the situation more complex.

my comments as;

1)      The comma should be added in the title a “…..field, and temperature…….”

2)      Line 140, “The mechanical strength test is carried 140 out at the temperature of 20℃± 10K”. the unit of the temperature should be the same.

3)      Line 291, “Fit the moisture absorption test data of composite cross arm mandrel based on Fick diffusion model [10], and calculate the diffusion coefficient of each sample, The results are shown in Figure 10.” The parameters in Fig. 10 should be defined and discussed.

4)      Line 292, “and calculate the diffusion coefficient of each sample, The results are shown in Figure 10”. This sentence needs to be revised.

5)      Line 387, “In this paper, the empirical formula of reaction rate varying with temperature, Arrhenius equation and reaction kinetic model are used to calculate the activation energy of materials before and after aging [13-16].” The way for calculating the activation energy should be included in the text.

Line 481, Page number?

Author Response

(The authors gave the same response as above.)

Reviewer 3 Report

The manuscript satisfactorily covered key aspects of this field of study with acceptable scientific evidence. just some minor errors that need correction:

- Lines 291-93,  .....Fit the moisture absorption test data of composite cross arm mandrel based on Fick diffusion model [10], and calculate the diffusion coefficient of each sample, The results are shown in Figure 10.

-Lines 414-15,  .... During the aging process of 10kV Composite cross arm, it is affected by many factors, such as voltage, rain, high temperature, humidity, salt fog, ultraviolet light and mechanical fatigue.

The overall quality of English used was very good.

Author Response

(The authors gave the same response as above.)

Reviewer 4 Report

The Article is devoted to the experimental study of the aging of the traverse composite mandrel under the influence of various factors, such as stress, rain, temperature, humidity, salt fog, ultraviolet radiation and mechanical stress for a long time. The authors present the results of studying the aging of the traverse composite material by various methods, in particular, hydrophobic properties testing, mechanical properties testing, performance testing, moisture absorption rate, electrical testing, TGA, SEM analysis and 3D-CT non-destructive testing. The main conclusions of the Article are confirmed by the presented experimental results. The Article is of interest to the journal. The following comments are made on the text of the article:

1. The text of the Article contains a large number of stylistic errors, some of which are shown in the attached pdf file.

2. The Abstract and the beginning of the article should be more compact and consistent.

3. Authors must check the links. In particular, reference [10] is incorrect.

4. Presented average values (page 8) should be accompanied by an indication of their errors (see comment in pdf file).

5. In some cases, figure captions should be written in more detailed.

6. Authors should check what is shown in the Figures.

According to the reviewer, the Article can be published after minor corrections.

Extensive editing of English language required

Author Response

(The authors gave the same response as above.)

Round 2

Reviewer 1 Report

After revision on the newest version of the manuscript, there are non further comments because improvements were carried out based on comments.

Please check capitalization letter in: keywords, page 2 line 49 (Composite), page 5 line 190 (Nova nano sem450 ),page 6 line 206 (Advantage).

Chek on page 7 line 225 the presence of letter B (3.2. B. Hydrophobic characteristics of outer sheath of composite cross arm after aging )

Check on Page 9 line 303, the use of coma at the end of phrase.